# Nonconventional Yeasts Engineered Using the CRISPR-Cas System as Emerging Microbial Cell Factories

**Jongbeom Park [1], In Jung Kim [2],\* and Soo Rin Kim [1,3],\***

[1] School of Food Science and Biotechnology, Kyungpook National University, Daegu 41566, Republic of Korea
[2] Department of Applied Biosciences, Graduate School, Kyungpook National University, Daegu 41566, Republic of Korea
[3] Research Institute of Tailored Food Technology, Kyungpook National University, Daegu 41566, Republic of Korea
\* Correspondence: ij0308@knu.ac.kr (I.J.K.); soorinkim@knu.ac.kr (S.R.K.); Tel.: +82-53-950-6654 (I.J.K.); +82-53-950-7769 (S.R.K.)

**Abstract:** Because the petroleum-based chemical synthesis of industrial products causes serious environmental and societal issues, biotechnological production using microorganisms is an alternative approach to achieve a more sustainable economy. In particular, the yeast *Saccharomyces cerevisiae* is widely used as a microbial cell factory to produce biofuels and valuable biomaterials. However, product profiles are often restricted due to the Crabtree-positive nature of *S. cerevisiae*, and ethanol production from lignocellulose is possibly enhanced by developing alternative stress-resistant microbial platforms. With desirable metabolic pathways and regulation in addition to strong resistance to diverse stress factors, nonconventional yeasts (NCY) may be considered an alternative microbial platform for industrial uses. Irrespective of their high industrial value, the lack of genetic information and useful gene editing tools makes it challenging to develop metabolic engineering-guided scaled-up applications using yeasts. The recently developed clustered regularly interspaced short palindromic repeats (CRISPR)-associated protein (Cas) system is a powerful gene editing tool for NCYs. This review describes the current status of and recent advances in promising NCYs in terms of industrial and biotechnological applications, highlighting CRISPR-Cas9 system-based metabolic engineering strategies. This will serve as a basis for the development of novel yeast applications.

**Keywords:** nonconventional yeast; genome editing; metabolic engineering; CRISPR-Cas9 system

## 1. Introduction

Petroleum-derived chemical production has detrimental effects on the environment and exhibits industrial noncompatibility associated with cost-effectiveness due to multiple labor-intensive processes [1]. To overcome this problem, biotransformation using microorganisms can be considered an alternative approach [1,2], which has the advantage of rapid growth rate and easy cultivation of microorganisms under laboratory conditions [3]. In particular, eukaryotic yeasts represent robust microbial cell factories owing to their simple structure and ability to grow on various substrates, as well as the relatively simple gene editing techniques used to manipulate their genomes [4].

*Saccharomyces cerevisiae* is the most widely used eukaryotic platform in bioprocesses [5,6]. *S. cerevisiae* has long been a model organism for fundamental biological research and industrial applications because of its ease of handling and safety as a generally recognized as safe (GRAS) strain [7]. Moreover, its genetics are well understood, and tools for manipulating it are well established; thus, numerous specialized strains and plasmids are available. However, as a Crabtree-positive organism, the carbon flux in *S. cerevisiae* is mainly directed toward the ethanol fermentation pathway. This preference for ethanol production often limits its utilization as a host when nonethanol products are to be synthesized. Therefore, other suitable yeasts, known as nonconventional yeasts (NCY), should be considered as alternative hosts.

Approximately 1500 NCY species have been identified to date [8], each of which exhibits unique genetics, physiology, and characteristics. They often have excellent potential for industrial uses that are not feasible with *S. cerevisiae*. Thus, NCY can be considered as promising eukaryotic hosts alternative to *S. cerevisiae* to overcome or improve its limitations. Specifically, NCYs that are Crabtree-negative can diversify the profile of industrially useful products. In addition, a much higher capacity of NCYs for pentose phosphate pathway relative to *S. cerevisiae* is also advantageous feature when synthesizing products using NCYs as the cell factory by increasing the available pool of cofactors and precursors [9]. Moreover, a high tolerance against multiple stress factors, such as heat, low pH, and salt, can extend yeasts' utility. Advancements in genetic and metabolic engineering technologies will facilitate NCY-based scaled-up bioprocesses [8,10].

Despite their beneficial traits, genetic information and manipulation tools for many NCYs are lacking compared with those for *S. cerevisiae* [11]. As most organisms have not been thoroughly analyzed for safety and lack genome sequencing data with the limited information on the exact gene loci and protein functions, it is challenging to develop and apply suitable gene editing tools, together with establishing transformation protocols and selectable marker genes [11].

The clustered regularly interspaced short palindromic repeats (CRISPR)-associated protein (Cas) system is an adaptive immune system of bacteria and archaea that protects them from invasion by foreign genetic elements [12]. As a gene editing tool, the CRISPR-Cas system is based on a simple single guide RNA (sgRNA)/DNA hybrid that recognizes specific target DNA, providing a simple-to-design methodology, which has more sophisticated, accurate, and cheaper gene editing capabilities compared with traditional methods like zinc finger nucleases (ZFNs) and transcription activator-like effector nucleases (TALENs) [13]. Thus, this revolutionary CRISPR-Cas9 system has been applied to various yeast strains, such as *S. cerevisiae*, *Pichia pastoris*, *Kluyveromyces marxianus*, and *Yarrowia lipolytica*; among these, most studies have been conducted on *S. cerevisiae* [14,15]. Although numerous studies have reported the CRISPR-guided metabolic engineering of *S. cerevisiae*, only few studies have explored the applicability of this system in NCYs [16]. To achieve commercial scale bioproduction using NCYs as cell factories, it is necessary to develop highly efficient and convenient engineered strains [10,17].

In this review, we provide an overview of NCYs with excellent potential for industrial applications. In particular, the biotechnological applications of engineered NCYs using the CRISPR-Cas9 system are highlighted. The advances and challenges of CRISPR–Cas9-mediated biotechnology for NCYs are also discussed.

## 2. Industrial Value of NCYs

NCYs have several advantages over *S. cerevisiae* from an industrial viewpoint [18]. *S. cerevisiae* is often directed toward ethanol synthesis (due to its Crabtree-positive effect), restricting product diversification. In contrast, NCYs may have desired metabolic pathways, enabling product profile expansion. The ability to resist various stresses is a key benefit in industrial bioprocesses. For example, ethanol production from lignocellulose can be enhanced by developing alternative microbial platforms that are highly resistant to inhibitors. NCYs often exhibit strong resistance to various stresses, such as heat, acid, and high sugar concentrations, as environmental adaptations. Another key advantage of NCYs is their ability to utilize a wide range of carbon sources [8,19]. Additionally, many NCYs can exist in both haploid and diploid types like *S. cerevisiae*, and sexual reproduction is possible. Therefore, NCY strains with a desired ploidy can be developed through mating depending on the purpose. For example, diploid *P. pastoris* strains having a higher stability than its haploid form were constructed through a well-designed mating process for production of proteins [20], or various auxotrophic *K. marxianus* libraries were constructed using mating and dissection [21]. In this section, the industrial potentials of five promising NCYs are described (Figure 1).

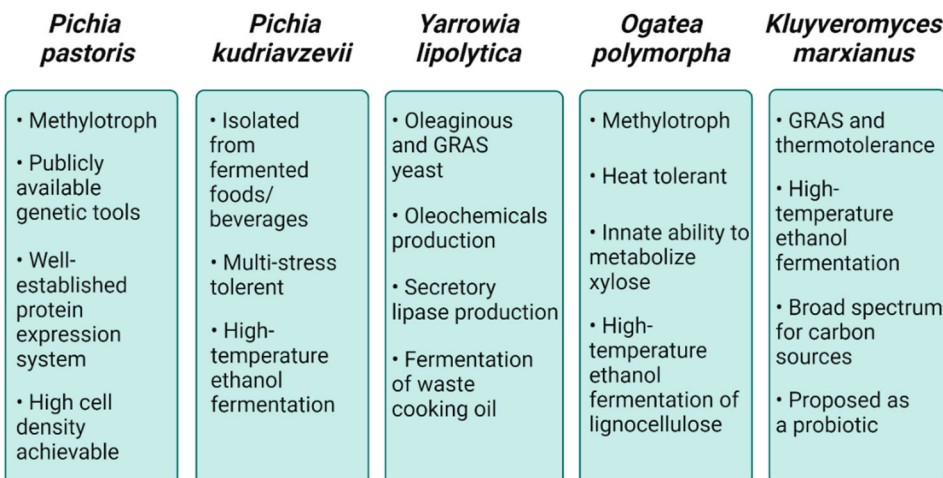

**Figure 1.** Properties and industrial values of promising NCYs.

### 2.1. Pichia pastoris

*P. pastoris* is a methylotrophic yeast with a developed peroxisomal system responsible for compartmentalized methanol metabolism [22]. Its gene expression system, together with its secretory system, is well established, which enables easy genetic manipulation using a publicly available commercial kit. *P. pastoris* is mainly used for the production of pharmaceutical proteins and other industrial enzymes, because of simple post-translational modifications process [23,24]. In fact, hyperglycosylation of proteins in *S. cerevisiae* often causes an allergic reaction in human body; therefore, using it as an expression host for producing pharmaceuticals and medicinal proteins is often undesirable. *P. pastoris* exhibits superior growth rate and cell density compared to *S. cerevisiae*, and like *S. cerevisiae*, its protein expression is controlled under strong and tightly regulated methanol inducible promoters (AOX1). Thus, high cell density in addition to a high yield of recombinant proteins, produced either intracellular or extracellular, can be achieved, which may lead to increased biotransformation efficiency of whole cells [23,25].

### 2.2. Pichia kudriavzevii

*P. kudriavzevii* is a multi-stress tolerant yeast commonly found in fermented foods and beverages, such as *Nuruk*, which is a starter used for making Korean traditional alcoholic beverages and various sub-Saharan African indigenous foods [26–29]. Previous studies have isolated robust strains of *P. kudriavzevii* that can withstand multiple stress factors, such as high salt concentration, high temperature, and low pH. In particular, there are several advantages of using thermotolerant strains as production hosts for the ethanol industry. For *S. cerevisiae*, scaled-up ethanol production through simultaneous saccharification and fermentation is normally performed at 30 °C, above which growth and fermentation are repressed. Ethanol production at a high temperature is beneficial for reducing microbial contamination, as well as energy and water costs required to cool the fermentation system [30–32]. Indeed, thermotolerant *P. kudriavzevii* produce more ethanol at a higher temperature (44 °C) compared with *S. cerevisiae*. Furthermore, considering the information on genome sequence and genetic engineering tools, it is a potent host for various industrial metabolites, such as organic acids (e.g., succinic acid) and bioethanol [30,32–34].

### 2.3. Yarrowia lipolytica

As an oleaginous microorganism that accumulates lipids up to 20% of dry cell weight, *Y. lipolytica* is used for the industrial production of fatty acid-derived products [35]. *Y. lipolytica* is a GRAS organism and assimilates hydrophilic (e.g., glucose, glycerol, alcohols, and acetate) and hydrophobic substrates (e.g., fatty acids, triacylglycerols, and alkanes) [36]. With its high protein secretory capacity and lipophilicity, the organism has also been used

for the fermentation of waste cooking oil to achieve bioremediation and waste valorization. During this fermentation process, *Y. lipolytica* produces extracellular lipase, which subsequently generates free fatty acids, utilizable as a carbon source, from waste cooking oil. With the development of metabolic engineering, *Y. lipolytica* can potentially produce several other metabolites, such as organic acids [37], erythritol [38], and flavonoids [39].

*2.4. Ogataea polymorpha*

*O. polymorpha* is a methylotroph, an organism that utilizes C1 compounds, such as methanol, as its sole carbon source, and is one of the most heat-resistant yeasts. Because native *S. cerevisiae,* the most widely used workhorse for bioethanol production, is incapable of xylose fermentation and engineered *S. cerevisiae* to possess heterologous xylose metabolic pathway may suffer from high metabolic burden, a benefit of *O. polymorpha* is its innate ability to metabolize xylose, the second most abundant sugar of lignocellulosic biomass [40]. Thus, high-temperature (i.e., 45–50 °C) ethanol fermentation of lignocellulose hydrolysate mainly consisting of glucose and xylose is possible. As one of a few methylotrophic yeasts, their key enzymes involved in methanol metabolism are strongly induced by methanol present within membrane-bound peroxisomes, which enables a compartmentalized reaction. Based on this expression machinery, *O. polymorpha* is a useful expression host for producing heterologous and difficult-to-express proteins via establishing expression systems induced by methanol under the control of strong and tightly regulated promoters.

*2.5. Kluyveromyces marxianus*

*K. marxianus* is a GRAS and thermotolerant ethanol-producing species that can grow at temperatures up to 52 °C [41–43], enabling high-temperature ethanol fermentation. As a Crabtree-negative yeast, this species is also advantageous for synthesizing non-ethanol products. Like *K. lactis* [44], *K. marxianus* has the unique ability to assimilate lactose, which is not feasible with *S. cerevisiae* and other yeasts [44,45]. Owing to its high capacity to grow on a broad spectrum of cheap carbon sources, such as xylose, arabinose, galactose, lactose, pectin, inulin, hemicellulose hydrolysate, cheese whey, and molasses, this species is an excellent microbial source of enzymes, bioethanol, and food ingredients [46] for commercial-scale applications [47–49]. Additionally, *K. marxianus* can produce fructose and fructooligosaccharides, which are industrially pertinent foods and pharmaceutical ingredients through inulinase secretion [50]. Recently, *K. marxianus* has been proposed as a probiotic yeast due to its beneficial roles in the gut [46].

**3. Genetic Engineering Tools for NCYs**

Although the innate traits of NCYs are beneficial for industrial applications, the productivity is low and must be increased through metabolic engineering techniques. One basis for metabolic engineering is endogenous or heterologous gene expression using a host strain [51]. This can be accomplished through two approaches. The first approach is the development of an episomal plasmid expression system using a self-replicating vector and the second is genome integration [52]. For *S. cerevisiae*, episomal vector systems with high copy numbers have been highly developed through extensive optimization and are widely utilized relative to NCYs [53]. Further, plasmid-based expression systems have also been developed for NCYs [54,55]. For example, an autonomously replicating sequence (ARS)-based episomal vector has been developed for *P. pastoris*, providing higher transformation efficiency and lower interclonal variability compared with the classical integrative plasmid [56]. The lower interclonal variability of transformants obtained from ARS system could be highly related with the frequent occurring non-specific genome integration caused by the integrative vector [51]. However, such an episomal plasmid system suffers from a high cost and inefficiency arising from segregational instability, which requires its maintenance under selective pressure for stable protein expression. Moreover, a centromeric plasmid was engineered, in which various promoters were fused upstream of the centromere sequence to control the function of centromere in *Y. lipolytica*.

Such an approach successfully led to the significant improvement in the plasmid copy number, and in turn, protein expression levels (80%) [54].

From an industrial viewpoint, genome integration is the preferred method because it leads to more homogeneous expression levels and, upon integration, expression cassettes are stably inherited with the respective chromosome. To achieve more sophisticated gene editing and higher integration efficiency, the double-stranded sequence at the desired site must be broken. Nuclease-based tools, such as ZFNs and TALENs, can be used for the genetic engineering of NCYs [57,58]. However, these methods are outdated with the emergence of CRISPR-Cas9 system. The Cre–lox system, which deletes the sequences between direct repeats of two loxP sites by catalyzing their recombination, leaving a single loxP site behind in the genome [59], is also a useful technology for producing knockout mutants without leaving a marker; however, some scars are left in the genome, or unwanted recombination may occur [10]. CRISPR-Cas9 system, established and predominantly used over the past decade, is a powerful gene editing tool that overcomes the problems associated with conventional tools [60].

Contrary to ZFN and TALEN-assisted approaches in which nucleases play a dual role in recognition and cleavage, the CRISPR-Cas9 system relies on designed sgRNAs that are specific to a target gene, and after the recognition, Cas9 cleaves the targeted site. This is an advanced genetic engineering tool that allows a simple-to-design plasmid construction for sgRNA expression, as well as rapid and precise cell programming [10].

## 4. CRISPR-Cas System-Guided Metabolic Engineering in NCYs

### 4.1. CRISPR-Cas System: Classification, Components, and Mechanism

The CRISPR-Cas system, comprising a DNA array and associated proteins, is a widely distributed RNA-based adaptive defense mechanism against viruses in bacteria and archaea [61]. Based on the Cas protein effector, CRISPR-Cas systems are mainly categorized into two classes: class I (type I, III, and IV) with a multi-subunit Cas complex and class II (type II, V, and VI) with a single Cas protein [62,63]. Depending on the type and mechanism of Cas endonucleases, each type can be further classified into several subtypes.

Among these, the type II CRISPR-Cas9 system uses a single Cas9 protein and is most commonly used for genetic and metabolic engineering because of its simple structure. In particular, the most widely used CRISPR system is composed of a Cas9 protein derived from *Streptococcus pyogenes* [64]. The CRISPR-Cas9 system consists of two integral components, Cas9 and sgRNA; Cas9 is divided into two domains: a recognition lobe (REC1 and REC2) and nuclease lobe (NUC). Further, the sgRNA is a fusion construct composed of the two RNA types, CRISPR RNA (crRNA) and transactivation crRNA (tracrRNA), connected by a linker sequence (Figure 2) [65]. sgRNA is bound by the REC lobe of Cas9 and REC lobe controls the conformational alteration of the catalytic core of Cas9 (i.e., HNH motif) [66]. The NUC lobe comprises the following three domains: (1) HNH, (2) RuvC cleaving target and nontarget single-stranded DNAs, and (3) protospacer adjacent motif (PAM)-binding domain that scans PAM sequence through weak or transient interactions before forming a stable hybrid between target sequence-sgRNA to perform the specific cleavage.

The process of Cas9-mediated genome editing involves recognition, cleavage, and repair [67]. First, the PAM sequence (i.e., 5′-NGG-3′) is scanned for probing the target for cleavage and subsequently, the target sequence is recognized by the sgRNA bound to REC lobe of Cas9 [64,67]. After the sgRNA forms a duplex with its complementary DNA, the target and nontarget single-stranded DNA sequences located at 3-bp upstream of the PAM sequence are cleaved by HNH and RuvC nucleases, respectively [65]. Finally, the generated double-strand breaks (DSBs) undergo DNA repair; the nucleotide sequence of the cleaved site can be modified through the endogenous DNA repair machinery. Two main repair systems in eukaryotes are based on the homology-directed repair (HDR) and non-homologous end joining (NHEJ) pathways [68]. The predominant pathway in eukaryotes is NHEJ in which random insertions and deletions (indels) of a small number of nucleotides are introduced at the broken ends. Without using template DNA, this

process is prone to error and often causes undesirable genetic modifications. In contrast, the HDR pathway uses a homologous DNA template for DSB repair that can be exogenously provided as single- or double-stranded DNA carried as a plasmid or PCR product. The designed nucleotide sequence is integrated into the genome of a host strain via homologous recombination (HR) [69]. This template-based mechanism allows accurate genome editing, modification, and replacement.

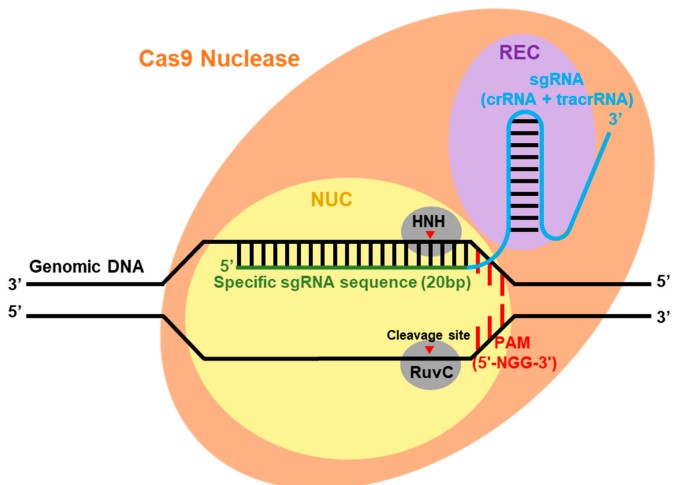

**Figure 2.** The CRISPR-Cas9 system: Cas9 structure and mechanism. Abbreviations: sgRNA, single guide RNA; crRNA, CRISPR RNA; tracrRNA, transactivation crRNA; PAM, protospacer adjacent motif; REC, recognition lobe; NUC, nuclease lobe.

### 4.2. Challenges and Strategies of CRISPR-Cas9-Guided Genome Editing in NCYs

As demonstrated in Section 4.1, the CRISPR-Cas9 system has emerged as the most powerful tool for the gene editing of yeast species for biotechnological applications due to its simple design, precise recognition and targeted generation of double strand breaks, and no remnant of antibiotic selection markers [70]. Here, we summarize the CRISPR-Cas9 system applied to NCYs (Table 1).

**Table 1.** Comparison of CRISPR-Cas9 system for NCYs.

| Strain | sgRNA Promoter | Plasmid (Backbone) | Cas9 Promoter | Editing Efficiency [1] (%) | Reference |
|---|---|---|---|---|---|
| *Y. lipolytica* | *SCR1′-tRNA* [2] *TEFin* (Pol II) [3] | pCRISPRyl pCASyl pGGA | *TEF1* | 0–68.9 | [71–74] |
| *O. polymorpha* | *ScSNR52* (Pol III) *ScTDH3* (Pol III) | pCRCT pYTK079 | *ScTEF1* *AaTEF* | 1–75 | [75,76] |
| *P. pastoris* | *HTX1* (Pol II) [3] *PFK300* (Pol II) [3] *LAT1* (Pol II) [3] | pPpT4 p414 BB3cH | *HTX1* *GAP* | 75–93.8 | [77–79] |
| *K. marxianus* | *ScTDH3* (Pol II) [3] *RPR1′-tRNA* (Pol III) [2] | pYTK079 pIW601 | *AaTEF* *1 ScTEF1* | 10–82 | [76,80] |
| *P. kudriavzevii* | *RPR1* (Pol III) *RPR1′-tRNA* (Pol III) [2] | pRS416 pRS415 pCast | *TEF1* | 64 | [30,81,82] |

[1] Defined as transformation efficiency (i.e., positive colony number/total colony number); [2] synthetic hybrid promoters; [3] self-cleaving ribozyme system was used.

In contrast to the homologous DNA template-based HDR system adopted by *S. cerevisiae*, most NCYs rely on the NHEJ pathway. NHEJ-dominant DNA repair system is the major challenge of CRISPR-Cas9-guided genome editing for NCYs. In particular, NHEJ is an evolutionarily conserved pathway that directly rejoins the broken ends of DNA without

a template [83–85]. This recovery system can cause unwanted frameshift indels, which is an obstacle to integrating donor DNA into genomes [17]. To overcome this limitation, a strategy to knockout NHEJ-related genes was devised that was capable of strongly reducing NHEJ, causing remaining recombinants to have a higher percentage of the desired HDR [86]. Through knockout of *Ku70/Ku80*, *Dnl4/Lif1*, *Nej1*, and *Mre11/Rad50/Xrs2* (*MRX*), studies have shown that the NHEJ pathway was inhibited but gene targeting efficiency was increased through the induced HDR pathway [87–89]. Although the stability of strains associated with growth and ultraviolet sensitivity needs to be improved [89], these NHEJ knockout approaches serve as a basis for CRISPR-guided genetic engineering of NCYs. Conversely, there is a strategy to increase genome integration efficiency through overexpression of genes involved in the HDR pathway [90]. For example, RAD51 or 52, a recombinase involved in DSB repair, is the central enzyme of HDR process, and its overexpression is suggested as a methodology to induce HDR pathway. Such an approach has either enhanced or inhibited HDR efficiency in earlier studies [91]. While a study revealed that RAD51 overexpression inhibited HDR process of DSBs in yeast (i.e., *S. cerevisiae*) [91], there are also examples with *P. pastoris* showing that heterologous or homologous expression of RAD52 increases HR efficiency [92]. A similar phenomenon was also observed with *Y. lipolytica* [93].

The CRISPR-Cas9 system requires the optimal expression of sgRNA and Cas9 that are tailored to the individual organism. sgRNA expression is another major challenge, and its efficiency relies on the type of RNA promoter and polymerase. According to a previous study, sgRNA expression was observed under the control of the standard RNA polymerase III promoter in *S. cerevisiae*, but expression of gRNA using the same promoter was at the low level in NCYs, including *P. pastoris* [94]. Furthermore, RNA polymerase II, a polymerase responsible for most protein-coding mRNA synthesis, is not applicable for sgRNA transcription because the transcripts synthesized by this RNA polymerase cause significant alterations, such as polyadenylation, at the transcript termini, which are undesirable for sgRNA expression [95]. To overcome these limitations, Gao et al. developed a chimeric gene system known as ribozyme-sgRNA-ribozyme, through which the RNA transcript is self-cleaved because of the ribozymal nuclease activity to release the sgRNA [95,96]. This system functions well with appropriate promoters using the classical RNA polymerase (e.g., RNA polymerase II), which serves as an efficient tool to increase functional sgRNA expression, even in NCYs (Table 1).

Apart from those mentioned above, challenges, such as off-target effects causing unwanted target cleavage [97] and restriction of NGG PAM motifs [98] remain, and further improvements are needed for industrial application of NCYs. To be specific, variants of Cas9 through protein engineering (for recognition of NAG and NGA PAMs) [99] and PAM sequence can be generated to achieve the more specific target cleavage. Additionally, an off-target effect could be solved by the well-designed sgRNA using *in silico* program [100].

### 4.3. Biotechnological Application of CRISPR-Cas9-Introduced NCYs

#### 4.3.1. Secondary Plant Metabolites

Plant-derived secondary metabolites are organic compounds with complex molecular structures that exhibit beneficial bioactivity. Flavonoids, terpenes, saponins, alkaloids, and sterols are representative examples. These exhibit antiobesity, anticancer, antioxidant, and antiaging effects [101,102] and can be used in various industries, such as pharmaceuticals, foods, and cosmetics [103,104]. The classical production of these compounds mainly depends on extractive and chemical methods, which are neither environmentally friendly nor economically feasible [105]. Microbial biosynthesis is a promising alternative to overcome the problems associated with current production methods [106]. *Y. lipolytica* is a favorable host strain for the biosynthesis of secondary plant metabolites, especially flavonoids, as it produces large amounts of acetyl-CoA and other flavonoid precursors through the mevalonate (MVA) pathway. With the application of CRISPR-Cas9 technology for genomic

integration in *Y. lipolytica*, targeted and marker-free strains have been developed in an easy, simple, and precise manner, thus contributing to the current research [107].

Flavonoids, such as naringenin and resveratrol, are a class of secondary plant metabolites derived from phenylpropanoids [108]; they are biologically synthesized as follows (Figure 3). From phenylalanine and tyrosine, 4-coumaric acid is produced through deamination, which is then converted into 4-coumaroyl CoA by 4-coumaroyl-CoA ligase (4CL) [103]. Next, it is combined with three molecules of malonyl-CoA via polyketide synthase to form flavonoids. In yeasts lacking the aforementioned pathway, the heterologous expression of tyrosine amino lyase (TAL), 4CL, and chalcone synthase (CHS) is used as the primary strategy to produce flavonoid precursors [103,109]. In particular, an engineered *Y. lipolytica* strain in which the xylose metabolism and naringenin biosynthesis pathways were introduced produced 715.3 mg/L naringenin from a mixture of glucose and xylose as the carbon source (Table 2) [109]. Additionally, the exogenous expression of the resveratrol biosynthetic pathway and mutations in 3-deoxy-D-arabino-heptulosonic acid 7-phosphate synthase (ARO4) and chorismate mutase (ARO7) lowered the sensitivity to tyrosine-induced feedback inhibition, thereby increasing resveratrol production to 12.4 g/L [103].

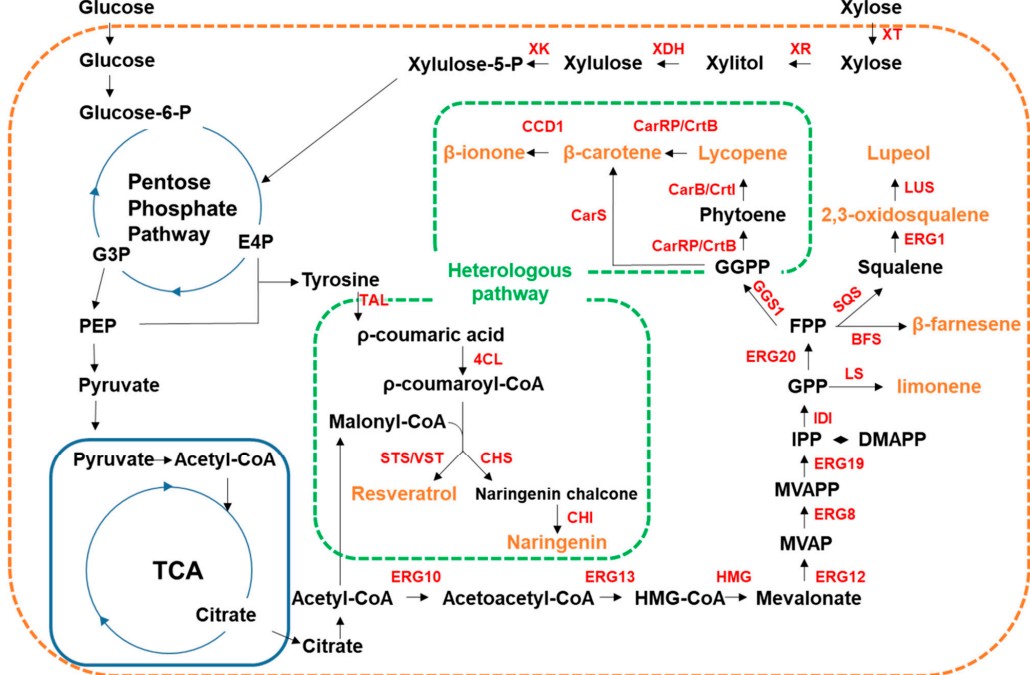

**Figure 3.** Engineered metabolic pathways for the production of secondary plant metabolites, such as flavonoids and carotenoids, in NCYs. Abbreviations: glucose-6-P, glucose-6-phosphate; xylulose-5-P, xylulose-5-phosphate; E4P, erythrose 4-phosphate; G3P, glyceraldehyde-3-phosphate; PEP, phosphoenolpyruvic acid; acetyl-CoA, acetyl coenzyme A; HMG-CoA, 3-hydroxy-3-methylglutaryl-CoA; MVAP, mevalonate-5-phosphate; MVAPP, mevalonate pyrophosphate; IPP, isopentenyl pyrophosphate; DMAPP, dimethylallyl pyrophosphate; GPP, geranyl pyrophosphate; FPP, farnesyl pyrophosphate; GGPP, geranylgeranyl pyrophosphate; TAL, tyrosine ammonia-lyase; 4CL, 4-coumarate-coa ligase; STS, stilbene synthase; VST, resveratrol synthase; CHS, chalcone synthase; CHI, chalcone isomerase; ERG10, acetyl-CoA C-acetyltransferase; ERG13, HMG-CoA synthase; HMG, HMG-CoA reductase; ERG12, mevalonate kinase; ERG8, phosphomevalonate kinase; ERG19, mevalonate pyrophosphate decarboxylase; IDI, isopentenyl diphosphate isomerase; LS, limonene synthase; ERG20, farnesyl pyrophosphate synthase; BFS, β-farnesene synthase; GGS1, geranylgeranyl diphosphate synthase; SQS, squalene synthase; ERG1, squalene monooxygenase; LUS, lupeol synthase; CarRP/CrtB, phytoene synthase/lycopene cyclase; CarB/CrtI, phytoene dehydrogenase; CarS, multi-functional carotene synthase; CCD1, carotenoid cleavage dioxygenase; XK, xylulokinase; XDH, xylitol dehydrogenase; XR, xylose reductase; XT, xylose transporter.

Carotenoids are tetraterpenes containing 40 carbon atoms and natural antioxidants. Carotenoids derived from vegetable and fruits, called β-carotene, β-cryptoxanthin, and α-carotene, are regarded as great dietary sources of provitamin A, which is converted in the human intestine into vitamin A, demonstrating their high industrial value [110]. *Y. lipolytica* natively produces geranylgeranyl pyrophosphate (GGPP) via the MVA pathway, which can be transformed into carotenoids through the introduction of heterologous genes involved in downstream pathways (Figure 3). In previous studies, lycopene and β-carotenoids were generated from GGPP via the introduction of phytoene dehydrogenase and phytoene synthase/lycopene cyclase from *Mucor circinelloides* or CrtYB and CrtI from *Xanthophyllomyces dendrorhous* [111,112]. Additionally, the production of β-carotene was increased to 4.5 g/L by enhancing the supply of precursors (i.e., GGPP) through the overexpression of MVA metabolic pathway genes, such as 3-hydroxy-3-methylglutaryl-CoA reductase [73].

Industrially useful terpenes, such as monoterpene and sesquiterpene, which are utilized in food, cosmetics, medicine, and next-generation jet fuel, have also been synthesized via the metabolic engineering of NCYs [113].

**Table 2.** Production of secondary plant metabolites using CRISPR-Cas9 in NCYs.

| Strain | Target Genes | | Product | Reference |
|---|---|---|---|---|
| | Endogenous Gene Editing | Heterologous Gene Editing | | |
| *Y. lipolytica* | HMG1, GGS1 | crtE (*Pantoea ananatis*), crtI (*P. ananatis*), crtB (*P. ananatis*) | Lycopene 3.38 mg/g DCW [2] | [71] |
| | GGS1 | carB (*Mucor circinelloides*), carRP (*M. circinelloides*), | β-carotene 4.8 mg/g DCW | [72] |
| | GGS1, ERG13, HMG | carB (*M. circinelloides*), carRP (*M. circinelloides*) | β-carotene 4.5 g/L | [73] |
| | GGS1, HMG1, ERG8, ERG10, ERG12, ERG13, ERG20, ERG19, IDI | carB (*M. circinelloides*), carRP (*M. circinelloides*), CCD1 (*Petunia hybrid*), PK (*Bifidobacterium bifidum*), PTA (*Bacillus subtilis*) | β-ionone 358.4 mg/L 0.98 g/L (fed-batch) | [112] |
| | XK, HMG1 [1], ERG12 [1] | LS (*Agastache rugosa*) [1], NDPS (*Solanum lycopersicum*) [1], XR (*Scheffersomyces stipitis*), XDH (*S. stipitis*) | Limonene 20.57 mg/L | [113] |
| | HMG, ERG12, IDI, ERG20, SQS | BFS (*Artemisia annua*), LS (*Citrus limon*), LS (*Perilla frutescens*), CnVS (*Callitropsis nootkatensis*), crtI (*Xanthophyllomyces dendrorhous*), crtYB (*X. dendrohous*), acs (*Salmonella enterica*) | β-farnesene 955 mg/L Limonene 35.9 mg/L Valencene 113.9 mg/L Squalene 402.4 mg/L β-carotene 164 mg/L 2,3-oxidosqualene 22 mg/L | [111] |
| | HMG1, ERG1, ERG9, OLE1, PAH1, DGK1 | LUS (*Ricinus communis*) | Lupeol 441.72 mg/L | [114] |
| | GGS1 | carS (*Schizochytrium* sp.) | β-carotene 0.41 mg/g DCW | [74] |
| | XT [1], XR [1], XDH [1], XKS [1] | TAL (*Rhodotorula glutinis*), 4CL (*Arabidopsis thaliana*), CHS (*A. thaliana*), CHI (*A. thaliana*) | Naringenin 715.3 mg/L | [109] |
| | ARO4, ARO7 | TAL (*Flavobacterium johnsoniae*), VST (*Vitis vinifera*), 4CL (*A. thaliana*) | Resveratrol 12.4 g/L | [103] |
| *O. polymorpha* | - | TAL (*Herpetosiphon aurantiacus*), STS (*V. vinifera*), 4CL (*A. thaliana*) | Resveratrol 97.23 mg/L | [75] |

[1] Plasmid-based engineering; [2] DCW: dry cell weight.

### 4.3.2. Other Industrial Products

In addition to secondary plant metabolites, various industrially useful products have been synthesized using NCYs engineered with the CRISPR-Cas9 system (Table 3).

For example, itaconic acid is a high value-added compound used to synthesize polymers and chemical intermediates, such as styrene and 2-methyl-1,4-butanediol [115]. Currently, *Aspergillus terreus* is used as the microbial host for the production of itaconic acid with a titer of 160 g/L; however, it has some disadvantages in terms of its pathogenic potential (biosafety level 2), heterogeneous fermentation (as the filamentous fungi), the resultant increase in viscosity of broth, difficult genetic engineering, and high sensitivity to shear stress, etc. [116]. Therefore, there is an increasing demand for an alternative host. Itaconic acid can be produced via cis-aconic acid decarboxylase (CAD), an enzyme that decarbonizes cis-carbonate, which is an intermediate of the tricarboxylic acid (TCA) cycle in yeasts [115]. Studies have focused on the heterologous expression of CAD from *A. terreus* [115,117]. The biosynthesis of itaconic acid using *Y. lipolytica* and *P. kudriavzevi* with biosafety, which have low pH resistance allowing for saving of downstream cost associated with neutralization, fast growth rates, and shear stress resistance, has been conducted [30,116]. One strategy to increase the amount of cis-aconitate secreted by mitochondria and transported into the cytosol is overexpressing the mitochondrial tricarboxylate transporter, which produced up to 22.03 g/L itaconic acid [116].

Lipid-derived oleochemicals have been increasingly synthesized by *Y. lipolytica*, a microbial host with a high potential to produce microbial oils that can replace vegetable oils to increase economic feasibility and reduce environmental pollution [118,119]. A titer of 25 g/L lipids was produced by knocking out phospholipase, an acyl-binding phospholipid hydrolase [119]. The rich malonyl-CoA pool in *Y. lipolytica* is also useful for fatty alcohol production. The heterologous expression of fatty acyl-CoA (FAR) from *Marinobacter aquaeolei* [98] in *Y. lipolytica* led to 5.75 g/L fatty alcohol [120].

*K. marxianus* can synthesize esters for industrial use as it contains a rich acetyl-CoA pool owing to its high growth rate and relevant endogenous enzymes, such as alcohol acetyltransferase and esterase [121–123]. With the development of CRISPR-based tools, it has become feasible to edit multiple target genes involved in related metabolic pathways. A previous study successfully increased the production of 2-phenylethanol to 850 mg/L in *K. marxianus* by balancing the precursors of shikimate and phenylalanine biosyntheses [48]. Additionally, 150 mg/L ethyl acetate was produced via the TCA cycle and knockdown of electron transport chain-related genes [123].

**Table 3.** Production of industrially useful products using CRISPR-Cas9 in NCYs.

| Strain | Target Genes | | Product | Reference |
|---|---|---|---|---|
| | **Endogenous Gene Editing** | **Heterologous Gene Editing** | | |
| *Y. lipolytica* | - | *CAD* (*Aspergillus terreus*), *mttA* (*A. terreus*) [1] | Itaconic acid 22.03 g/L | [116] |
| *P. kudriavzevii* | *ICD, mttA* [1] | *CAD* (*A. terreus*) [1] | Itaconic acid 1.23 g/L | [30] |
| *Y. lipolytica* | *SCT1, OLE1* | *FAR* (*M. aquaeolei*) | Fatty alcohol 5.75 g/L | [120] |
| *Y. lipolytica* | *PLA₂* | - | Lipid 25 g/L | [119] |
| *Y. lipolytica* | *AXP* | *celB* (*Pyrococcus furiosus*) | β-glycosidase 187.5 $\mu kat_{oNPGal}$/L [2] | [124] |
| *K. marxianus* | *ARO1, ARO2, ARO3, ARO4, ARO7, ARO8, ARO9, PHA2, TAL1, TKL1, RPE1, RKI1, LAC4* | *xfpk* (*Bifidobacterium breve*), *ppsA* (*Escherichia coli*), *pta* (*Salmonella enterica*) | 2-penylethanol 850 mg/L | [48] |
| *Y. lipolytica* | - | *FAP* (*Chlorella variabilis*) | Hydrocarbons 58.7 mg/L | [125] |
| *K. marxianus* | *ACO2b, SDH2, RIP1, MSS51* | - | Ethyl acetate 150 mg/L | [123] |

[1] Plasmid-based engineering; [2] The amount of enzyme was measured as enzyme activity assayed against *o*-nitrophenyl-β-galactopyranoside (oNPGal).

### 5. Future Perspectives

The current petroleum-based economy can be replaced by a sustainable bioeconomy with the increase in the development of modern biotechnology based on genetically and metabolically engineered microorganisms. The recent approval of CRISPR-Cas system-mediated genome editing for use in gene therapy clinical trials and relaxed regulation of the commercialization of genetically modified crops by the United States government will accelerate the expansion of applications of genome editing from bench to reality. This can be demonstrated by NCYs with intrinsic commercial value applicable to various industrial sectors. However, to overcome the technical difficulties associated with the genetic engineering of NCYs, careful tailoring of CRISPR-Cas systems in a given organism is required. The continual discovery of novel machinery and development of diverse variants of the CRISPR-Cas system are expected to promote this process. Advances in systems and synthetic biology approaches, in which predicted modeling data are combined with experimental data (i.e., product yield and metabolic flux), will further aid in the selection of suitable strains and optimize engineered pathways more efficiently for NCYs.

### 6. Conclusions

Each NCY possesses unique characteristics and a high potential for industrial use. Compared with the model yeast *S. cerevisiae*, the genetic and physiological characteristics of NCYs have not been fully elucidated, making it challenging to develop industrial applications using NCYs as microbial hosts. Although the introduction of the revolutionary CRISPR-Cas9 system has accelerated the targeted metabolic engineering of NCYs, error-prone genetic mutations emerging from native DNA repair machinery, depending on the inaccurate NHEJ pathway, are the primary obstacle. To overcome this issue, editing efficiency can be improved using strategies for knocking out NHEJ-related genes or overexpressing HR-related genes. Further improvement can be achieved by customized optimization of CRISPR-Cas9 systems specific to individual organisms.

**Author Contributions:** Conceptualization, J.P.; writing—original draft preparation, J.P.; writing—review and editing, I.J.K. and S.R.K.; supervision, S.R.K. All authors have read and agreed to the published version of the manuscript. **Funding:** This work was supported by a National Research Foundation of

Korea (NRF) grant funded by the Korean government (MSIT) (No. NRF-2022R1A2C1093074). This work was also supported by a National Research Foundation of Korea (NRF) grant funded by the Korean government (MSIT) (No. 2022R1I1A1A01072158).

**Conflicts of Interest:** The authors declare no conflict of interest.

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
