# Peer review of "Nonconventional Yeasts Engineered Using the CRISPR-Cas System as Emerging Microbial Cell Factories"

_fermentation, doi:10.3390/fermentation8110656_

Round 1

Reviewer 1 Report

The review at hand is intended to give an overview of the use of gene editing using CRISPR/Cas in yeasts of industrial importance other than Saccharomyces cerevisiae. Besides some misleading expressions and phrases, overall the review partially gives the impression of staying somewhat superficial and lacking some important details and critical discussions. I will therefore make some general points before going into the details of phrasing.

From Figure 1 the review seems to focus on five yeasts, namely two species of Pichia, Yarrowia lipolytica, Ogatea polymorpha, and Kluyveromyces marxianus. Since the success of S. cerevisiae as the conventional yeast is not only owed to its ease of genetic engineering, but also to its sexual life cycle allowing classical genetics, it is essential to mention the ploidy of the non-conventional species discussed, as well as if they can be subjected to crossings. I’m not an expert on these species, but at least for K. marxianus tetrad analyses were apparently reported (Yarimizu et al. Yeast 2013, doi: 10.1002/yea.2985). Is this still commonly used in the manipulation of industrial strains? – Anyway, this should be cited (especially in the context of the pitfalls introduced by knocking out the NHEJ system; see also below).

As for the description of the CRISPR/Cas9 system, there should be some more information than textbook knowledge. Methods like ZNFs and TALEN not really used by many people since years, and not being subject of this review, do not need a detailed explanation. Rather, one sentence mentioning that they are outdated suffices. This space can be better used to explain in more detail how the PAM sequences are scanned by transient interactions and stabilized only at the right target sites by hybridization with the sgRNA. Furthermore, a discussion of the importance of off-targets especially in the different yeast genomes should be added. This includes some quantitative data on the efficiency of the gene editing in each yeast species, with details on which vectors are used to introduce the system and which promoters are used for expression of Cas9 and the sgRNA coding sequence in each species.

Specific queries:

l. 101: “P. pasotris exhibits superiour growth performance and ...”. Does this mean it grows better/faster than S. cerevisiae? – If so, it can be stated in this more simple way. “Its protein expression is controlled ...” – This is not different from S. cerevisiae, where there are also tightly controlled and strong promoters avaiblable. It should also be stated that heterologous proteins are efficiently secreted, which is a major reason for using this yeast. As it stands, this fact is somewhat hidden in the discussion of glycosylation patterns, but only obvious to people familiar with this expression system.

l. 151: “K. marxianus has the unique ability to assimilate lactose ...” This is utterly wrong. In fact, K. lactis was the first yeast to be isolated from milk and is of biotechnological importance for the degradation of lactose. The authors may want to cite Alvarez-Cao et al. Microb. Cell Fact. 2018, doi: 10.1186/s12934-018-0988-6. In fact, it seems somewhat strange that K. lactis is only once mentioned in Table 2, while it is certainly one of the more common non-conventional yeasts (e.g. Rodicio and Heinisch, Yeast 2013, doi: 10.1002/yea.2954).

l. 154: The concept of “strain” should be corrected throughout the manuscript. The authors frequently speak of strains, when they actually mean the yeast species. I suspect that each species has a subset of different strains used for industrial purposes, i.e. there are different strains of K. marxianus in use, but the authors want to generally refer to the species K. marxianus.

l.177: “increases the stability of clones” is somewhat of an understatement. Although true, it may be better to state that upon integration the expression cassettes are stably inherited with the respective chromosome.

l. 186: The description of the Cre-loxP system needs to be improved. It does not “cleave and invert”, but it deletes the sequences between direct repeats of two loxP sites by catalyzing their recombination, leaving a single loxP site behind in the genome. Also, the process is neither especially expensive, nor time consuming. All it needs is a yeast transformation with a vector carrying the gene for the Cre recombinase. Time is only needed then to induce plasmid-loss, which can be forced by 5-FOA (e.g. Heik and Hegemann, 2011, doi 10.1007/978-1-61779-197-0_12).

l. 190: instead of “most recently developed” the authors may want to say that “within the last decade established and predominantly used ...”

l. 201: “bacteriophages and viruses” – this is redundant, since bacteriophages are defined as being viruses that specifically infect bacteria

l. 212: “sgRNA is composed of ...” the sgRNA is a fusion construct of the two RNA types connected by a linker sequence

l. 213: “The role of the REC lobe is to sense sgRNA ....” should be “sgRNA is bound by the REC lobe of Cas9

l. 217: “that mediates the specific cleavage” should be rephrased. The PAM sequence only very transiently interacts with the DNA and the specific cleavage is really mediated by the hybridization with the sgRNA, not by this weak interaction.

l. 224: Likewise, first recognition is not initiated by the binding of sgRNA. Rather, the PAM sequence is scanned, than the target sequence is recognized by the sgRNA bound to the REC lobe. Please correct phrasing.

l. 238: “is easily integrated” Easy or not doesn’t matter for the cell and makes no sense on molecular mechanisms. Please delete “easily”

l. 245: also delete “sophisticated” for similar reasons

l. 247: Delete first sentence of this paragraph, as it is highly redundant

l. 256: deletions of genes encoding components of NHEJ do not “suppress” random integration (this is a genetic term with other meaning) and do not “trigger the HR machinery”. Rather, NHEJ is strongly reduced, so that the remaining recombinants have a higher percentage of the desired homologous recombination.

l. 260: “repression” is also a standard genetic term for gene expression, i.e. transcription control; use NHEJ-less or NHEJ knockout approaches

l. 266: do the authors mean that these heterologous PolIII promoters do not work? – have homologous PolII promoters been tested? – If so, which ones? – Is the ribozyme system really exclusively employed in all of the non-conventional yeasts? – I find it hard to believe that not even in K. marxianus a simpler way has been tried. As stated in the general remarks, these systems should be explained in more detail for each yeast species discussed.

l. 289: Also as stated above, please give some critical discussion on off-target sites and pitfalls of the system in the different yeast species. Also, is the PAM sequence usually modified by the manipulations, so that repeated cleavage of the target site is excluded?

l. 306/307:  should read “40 carbon atoms”; what does “with industrial value of provitamin A” mean? – such as? – derived from? – producing? – please rephrase

Figure 3: here and in the text with discussion of the Crabtree effect and carbohydrate metabolism, the authors may want to state that the Pentose Phosphate Pathway can have a much higher capacity in non-conventional yeasts as opposed to S. cerevisiae (Bertels et al. Biomol. 2021, doi 10.3390/biom11050725).

Table 1: The headline should be above the table and not on the previous page. Also make sure that in the third column really all the genes have been introduced by CRISPR/Cas, i.e. by “gene editing” and none of them simply on plasmids, which would then be “gene expression”

l. 347: please give some more details; Aspergilli have been used since decades for the production of citric acid and also itaconic acid. Growth and production schemes have been well established. What exactly would be so much better using the yeast production system. Also in l. 355: 22.03 g/L compares to what productivity in A. terreus?

l. 362: like above, please check if all of the genes have been introduced by gene editing and none by conventional methods

Table 2: either expand the review to K. lactis as suggested above, or omit K. lactis from this table

l. 406: How would “overexpressing HR-related genes” work? – in the case of knocking out NHEJ, deletion of one component is sufficient to impede the error-prone repair. For increasing HR efficiency it would be necessary to improve a lot of components. I don’t see that this would be feasable.

Author Response

Reviewer #1

The review at hand is intended to give an overview of the use of gene editing using CRISPR/Cas in yeasts of industrial importance other than Saccharomyces cerevisiae. Besides some misleading expressions and phrases, overall the review partially gives the impression of staying somewhat superficial and lacking some important details and critical discussions. I will therefore make some general points before going into the details of phrasing.

Response: Thanks a lot for the reviewer’s critical point. We have tried to address all the points by the reviewer. Please, see the point-by-point responses as below.

Comment 1

From Figure 1 the review seems to focus on five yeasts, namely two species of Pichia, Yarrowia lipolytica, Ogatea polymorpha, and Kluyveromyces marxianus. Since the success of S. cerevisiae as the conventional yeast is not only owed to its ease of genetic engineering, but also to its sexual life cycle allowing classical genetics, it is essential to mention the ploidy of the non-conventional species discussed, as well as if they can be subjected to crossings. I’m not an expert on these species, but at least for K. marxianus tetrad analyses were apparently reported (Yarimizu et al. Yeast 2013, doi: 10.1002/yea.2985). Is this still commonly used in the manipulation of industrial strains? – Anyway, this should be cited (especially in the context of the pitfalls introduced by knocking out the NHEJ system; see also below).

Response: All the yeasts described in Figure 1 can exist in two types, haploid and diploid, and sexual reproduction is possible. As the reviewer suggested, the ploidy and possibility of crossing of the non-conventional species were discussed

Page 2, line 91-97: Also, many NCYs can exist in both haploid and diploid types like S. cerevisiae, and sexual reproduction is possible. Therefore, NCY strains with a desired ploidy can be developed through mating depending on the purpose. For example, diploid P. pastoris strains having a higher stability than its haploid form were constructed through a well-designed mating process for production of proteins [20], or various auxotrophic K. marxianus libraries were constructed using mating and dissection [21].

Comment 2

As for the description of the CRISPR/Cas9 system, there should be some more information than textbook knowledge. Methods like ZNFs and TALEN not really used by many people since years, and not being subject of this review, do not need a detailed explanation. Rather, one sentence mentioning that they are outdated suffices. This space can be better used to explain in more detail how the PAM sequences are scanned by transient interactions and stabilized only at the right target sites by hybridization with the sgRNA. Furthermore, a discussion of the importance of off-targets especially in the different yeast genomes should be added. This includes some quantitative data on the efficiency of the gene editing in each yeast species, with details on which vectors are used to introduce the system and which promoters are used for expression of Cas9 and the sgRNA coding sequence in each species.

Response: According to the reviewer, description for ZFNs and TALEN was simplified but a sentence that they are outdated was mentioned. Furthermore, “how the PAM sequences are scanned by transient interactions and stabilized only at the right target sites by hybridization with the sgRNA” was more detailed. Please, see our responses to Comment 11&12.

Regarding the importance of off-targets especially in the different yeast genomes: we have provided quantitative data on the efficiency of the gene editing in each yeast species as the Table 1 including the above-mentioned details. Please, also see our added text

Page 8, line 303-308: Apart from those mentioned above, challenges, such as off-target effects causing un-wanted target cleavage [97] and restriction of NGG PAM motifs [98] remain, and further improvements are needed for industrial application of NCYs. To be specific, variants of Cas9 through protein engineering (for recognition of NAG and NGA PAMs) [99] and PAM sequence can be generated to achieve the more specific target cleavage. Also, off-target ef-fect could be solved by the well-designed sgRNA using in silico program [100].

Page 6, line 260-261: Here we summarize the CRISPR-Cas9 system applied to NCYs (Table 1).

Specific queries:

Comment 3

  1. 101: “P. pasotris exhibits superiour growth performance and ...”. Does this mean it grows better/faster than S. cerevisiae? – If so, it can be stated in this more simple way. “Its protein expression is controlled ...” – This is not different from S. cerevisiae, where there are also tightly controlled and strong promoters avaiblable. It should also be stated that heterologous proteins are efficiently secreted, which is a major reason for using this yeast. As it stands, this fact is somewhat hidden in the discussion of glycosylation patterns, but only obvious to people familiar with this expression system.

Response: 1. “P. pastoris exhibits superiour growth performance and ...” was stated in a more specific way in the revised manuscript. 2. To avoid confusion from the sentence “Its protein expression is controlled ...”, “like S. cerevisiae” was added. 3. Also, the superior capacity of the species for secretory expression was mentioned.

Comment 4

  1. 151: “K. marxianus has the unique ability to assimilate lactose ...” This is utterly wrong. In fact, K. lactis was the first yeast to be isolated from milk and is of biotechnological importance for the degradation of lactose. The authors may want to cite Alvarez-Cao et al. Microb. Cell Fact. 2018, doi: 10.1186/s12934-018-0988-6. In fact, it seems somewhat strange that K. lactis is only once mentioned in Table 2, while it is certainly one of the more common non-conventional yeasts (e.g. Rodicio and Heinisch, Yeast 2013, doi: 10.1002/yea.2954).

Comment 5

  1. 154: The concept of “strain” should be corrected throughout the manuscript. The authors frequently speak of strains, when they actually mean the yeast species. I suspect that each species has a subset of different strains used for industrial purposes, i.e. there are different strains of K. marxianus in use, but the authors want to generally refer to the species K. marxianus.

Comment 6

l.177: “increases the stability of clones” is somewhat of an understatement. Although true, it may be better to state that upon integration the expression cassettes are stably inherited with the respective chromosome.

Page 5, line 194-196: From an industrial viewpoint, genome integration is the preferred method because it leads to more homogeneous expression levels and upon integration expression cassettes are stably inherited with the respective chromosome.

Comment 7

  1. 186: The description of the Cre-loxP system needs to be improved. It does not “cleave and invert”, but it deletes the sequences between direct repeats of two loxP sites by catalyzing their recombination, leaving a single loxP site behind in the genome. Also, the process is neither especially expensive, nor time consuming. All it needs is a yeast transformation with a vector carrying the gene for the Cre recombinase. Time is only needed then to induce plasmid-loss, which can be forced by 5-FOA (e.g. Heik and Hegemann, 2011, doi 10.1007/978-1-61779-197-0_12).

Response: The description of the Cre-loxP system has been improved based on the reviewer’s comment.

Page 5, line 200-204: The Cre–lox system, which deletes the sequences between direct repeats of two loxP sites by catalyzing their recombination, leaving a single loxP site behind in the genome [59], is also a useful technology for producing knockout mutants without leaving a marker; however, some scars are left in the genome, or unwanted recombination may occur [10].

Comment 8

  1. 190: instead of “most recently developed” the authors may want to say that “within the last decade established and predominantly used ...”

Page 5, line 204-206: CRISPR–Cas9 system, established and predominantly used over the past decadeis a powerful gene editing tool that overcomes the problems associated with conventional tools [60].

Comment 9

  1. 201: “bacteriophages and viruses” – this is redundant, since bacteriophages are defined as being viruses that specifically infect bacteria

Comment 10

  1. 212: “sgRNA is composed of ...” the sgRNA is a fusion construct of the two RNA types connected by a linker sequence

Page 5, line 226-227: Further, the sgRNA is a fusion construct composed of the two RNA types, CRISPR RNA (crRNA) and transactivation crRNA (tracrRNA), connected by a linker sequence

Comment 11

  1. 213: “The role of the REC lobe is to sense sgRNA ....” should be “sgRNA is bound by the REC lobe of Cas9

Comment 12

  1. 217: “that mediates the specific cleavage” should be rephrased. The PAM sequence only very transiently interacts with the DNA and the specific cleavage is really mediated by the hybridization with the sgRNA, not by this weak interaction.

Response: The relevant sentence was rephrased to present the meaning more correctly in the revised manuscript.

Page 5, line 231-233: protospacer adjacent motif (PAM)-binding domain that scans PAM sequence through weak or transient interactions before forming a stable hybrid between target sequence-sgRNA to perform the specific cleavage.

Comment 13

  1. 224: Likewise, first recognition is not initiated by the binding of sgRNA. Rather, the PAM sequence is scanned, than the target sequence is recognized by the sgRNA bound to the REC lobe. Please correct phrasing.

Response: According to the reviewer’s comment, the sentences were rearranged by changing the order of mentioned processes.

Page 6, line 239-241: First, the PAM sequence (i.e., 5’-NGG-3’) is scanned for probing the target for cleavage and subsequently, the target sequence is recognized by the sgRNA bound to REC lobe of Cas9 [64,67].

Comment 14

  1. 238: “is easily integrated” Easy or not doesn’t matter for the cell and makes no sense on molecular mechanisms. Please delete “easily”

Response: Deleted according to the reviewer’s comment.

Comment 15

  1. 245: also delete “sophisticated” for similar reasons

Response: Deleted according to the reviewer’s comment.

Comment 16

  1. 247: Delete first sentence of this paragraph, as it is highly redundant

Response: The relevant sentence was deleted according to the reviewer’s comment.

Comment 17

  1. 256: deletions of genes encoding components of NHEJ do not “suppress” random integration (this is a genetic term with other meaning) and do not “trigger the HR machinery”. Rather, NHEJ is strongly reduced, so that the remaining recombinants have a higher percentage of the desired homologous recombination.

Page 7, line 271-274: To overcome this limitation, a strategy to knockout NHEJ-related genes was devised that was capable of strongly reducing NHEJ, causing remaining recombinants to have a higher percentage of the desired HDR [86].

Comment 18

  1. 260: “repression” is also a standard genetic term for gene expression, i.e. transcription control; use NHEJ-less or NHEJ knockout approaches

Response: Thanks a lot for the reviewer’s good point. As commented, NHEJ repression approaches was changed to NHEJ knockout approaches.  

Page 7, line 274-279: Through knockout of Ku70/Ku80, Dnl4/Lif1, Nej1, and Mre11/Rad50/Xrs2 (MRX), studies have shown that the NHEJ pathway was inhibited but gene targeting efficiency was in-creased through the induced HDR pathway [87-89]. Although the stability of strains associated with growth and ultraviolet sensitivity needs to be improved [89], these NHEJ knockout approaches serve as a basis for CRISPR-guided genetic engineering of NCYs

Comment 19

  1. 266: do the authors mean that these heterologous PolIII promoters do not work? – have homologous PolII promoters been tested? – If so, which ones? – Is the ribozyme system really exclusively employed in all of the non-conventional yeasts? – I find it hard to believe that not even in K. marxianus a simpler way has been tried. As stated in the general remarks, these systems should be explained in more detail for each yeast species discussed.
  2. do the authors mean that these heterologous PolIII promoters do not work?

Response: Yes, for clarity, the manuscript was revised.

Page 8, line 292-293: but expression of gRNA using the same promoter was not successful in NCYs including P. pastoris

Comment 20

  1. 289: Also as stated above, please give some critical discussion on off-target sites and pitfalls of the system in the different yeast species. Also, is the PAM sequence usually modified by the manipulations, so that repeated cleavage of the target site is excluded?

Response: Following the reviewer’s suggestion, we have made discussion on off-target sites and pitfalls of the system in the different yeast species. However, it was difficult to elucidate them case-by-case due to the limited information. Nevertheless, we have added a Table 1 for comparison as requested by the reviewer, and hope this is enough. 

Regarding the second question: Unfortunately, we could not find the relevant articles with respect to NCYs. We presume that repetitive cutting of the target site can be solved by excluding the PAM sequence and the 20 bp guide sequence from the editing process. Since this is our speculation, we would not like to discuss this point for this manuscript to avoid any confusion. Instead, a sentence was added to mention this issue in the revised manuscript.

Page 8, line 303-305: Apart from those mentioned above, challenges, such as off-target effects causing unwanted target cleavage and restriction of 5’-NGG-3’ PAM motifs remain, and further improvements are needed for industrial application of NCYs. To be specific, variants of Cas9 through protein engineering (for recognition of NAG and NGA PAMs) [99] and PAM sequence can be generated to achieve the more specific target cleavage. Also, off-target effect could be solved by the well-designed sgRNA using in silico program

Comment 21

  1. 306/307:  should read “40 carbon atoms”; what does “with industrial value of provitamin A” mean? – such as? – derived from? – producing? – please rephrase

Comment 22

Figure 3: here and in the text with discussion of the Crabtree effect and carbohydrate metabolism, the authors may want to state that the Pentose Phosphate Pathway can have a much higher capacity in non-conventional yeasts as opposed to S. cerevisiae (Bertels et al. Biomol. 2021, doi 10.3390/biom11050725).

Comment 23

Table 1: The headline should be above the table and not on the previous page. Also make sure that in the third column really all the genes have been introduced by CRISPR/Cas, i.e. by “gene editing” and none of them simply on plasmids, which would then be “gene expression”

Response: The headline was correctly positioned. In Table 1, although CRISPR/Cas system had been all applied in the listed NCYs, some cases employed a combined approach using both CRISPR/Cas and plasmid systems. That is why the third column also included genes not introduced by CRISPR/Cas. To clarify, we added a footnote. Please, see our revised Table 2 and 3.  

Comment 24

  1. 347: please give some more details; Aspergilli have been used since decades for the production of citric acid and also itaconic acid. Growth and production schemes have been well established. What exactly would be so much better using the yeast production system. Also in l. 355: 22.03 g/L compares to what productivity in A. terreus?

Response: (1) As the reviewer commented, A. terreus is the well-established microbial platform for itaconic acid (IA) with 160 g/L titer. However, expansion of platform toolkit for IA is needed to overcome some limitations of A. terreus including its pathogenic potential, heterogeneous fermentation (as the filamentous fungi), and the resultant increase in viscosity of broth, etc. Some NCYs could provide advantages to overcome these. Manuscript was more detailed to emphasize the advantages of NCYs in comparison with A. terreus. (2) the titer in A. terreus is 160 g/L, much higher than 22.03 g/L. This was also presented.  

Page 11, line 381-386: Currently, Aspergillus terreus is used as the microbial host for the production of itaconic acid with a titer of 160 g/L; however, it has some disadvantages in terms of its pathogenic potential (biosafety level 2), heterogeneous fermentation (as the filamentous fungi), the resultant increase in viscosity of broth, difficult genetic engineering, and high sensitivity to shear stress, etc [116].

Page 11, line 390-392: The biosynthesis of itaconic acid using Y. lipolytica and P. kudriavzevi with biosafety, which have low pH resistance allowing for saving of downstream cost associated with neutralization, fast growth rates, and shear stress resistance, has been conducted [30,116].

Comment 25

  1. 362: like above, please check if all of the genes have been introduced by gene editing and none by conventional methods

Response: As pointed out, there were some genes engineered based on conventional methods. We have marked them with footnote. Please, see our revised Table 2 and Table 3. 

Comment 26

Table 2: either expand the review to K. lactis as suggested above, or omit K. lactis from this table

Response: We found there is no enough articles relevant to K. lactis-mediated product synthesis using CRISPR-Cas9 system. In addition, K. lactis does not belong to the subject of our manuscript (Section 2). Accordingly, K. lactis was removed from the table as requested by the reviewer. 

Comment 27

  1. 406: How would “overexpressing HR-related genes” work? – in the case of knocking out NHEJ, deletion of one component is sufficient to impede the error-prone repair. For increasing HR efficiency it would be necessary to improve a lot of components. I don’t see that this would be feasable.

Response: Thanks a lot for the reviewer’s good point. Although the reviewer’s point that HR process involves several enzymes, which makes it difficult to increase HR efficiency simply by overexpressing them is reasonable, there are successful examples showing overexpression of RAD51 or 52, the central enzyme involved in HR process, have actually led to the increase in HR efficiency in various organisms including non-conventional yeasts. In this regard, we have addressed this issue by discussing the above-mentioned articles in the revised manuscript.

Page 7, line 279-287: Conversely, there is a strategy to increase genome integration efficiency through overex-pression of genes involved in the HDR pathway [90]. For example, RAD51 or 52, a recom-binase involved in DSB repair, is the central enzyme of HDR process, and its overexpres-sion is suggested as a methodology to induce HDR pathway. Such an approach has either enhanced or inhibited HDR efficiency in earlier studies [91]. While a study revealed that RAD51 overexpression inhibited HDR process of DSBs in yeast (i.e., S. cerevisiae) [91], there are also examples with P. pastoris showing that heterologous or homologous expression of RAD52 increases HR efficiency [92]. A similar phenomenon was also observed with Y. lipolytica [93]. 

Reviewer 2 Report

Review on:

Nonconventional yeasts engineered using the CRISPR–Cas sys- 2 tem as emerging microbial cell factories

The review describes the utilization of CRISPR/Cas9 in non-conventional yeasts and catalogs the previously undertaken efforts.

As such this is a nice overview.

It lacks a bit the description of various Cas9 enzymes and their PAM sequences as there is a larger diversity that could be used to also overcome off-target effects.

The authors conclude that there is an obstacle in NHEJ and CRISPR use. Maybe a look at some papers describing the overexpression of RAD51 to circumvent this issue would help in brightening up their outlook.

“Moreover, a superior defensive system against multiple stress factors can extend yeasts’ utility.” - There is no such thing as a “superior defense system”- rephrase and include specific examples e.g. high tolerance against salt, pH, temperature, osmotic stress etc.

L57-59: I guess there are genome sequences for all these NCY. Otherwise CRISPR guide RNAs would be difficult to generate. Also these yeasts will all have safety level 1 status - even though they may not have been assigned GRAS. Actually, it is not challenging to develop suitable gene editing tools. What is challenging is to establish transformation protocols and selectable marker gnees to get going in the first place.

L66-67: what is meant by traditional methods? TALEN? Zn-Finger? Or simple homologous recombination? Be specific.

L71: use Non-conventional yeast once and then abbreviate as NCY. Please change through-out the manuscript.

Fig 1: tolerant; modify ‘strong heat resistance’ - this may apply to archaea, certainly not to yeasts.

2.1. rephrase ‘with a peroxisome system’ what is meant?

L113: it is saccharification

L125: non-pathogenic. And no: non-pathogenic does not equal GRAS. Be clear about the regulatory status.

L136+: S.cerevisiae may be incapable of xylose fermentation. Yet, there are 2nd generation biofuel yeasts that can do this job. Change and add appropriate references.

Delete: “The advantages of high-temperature fermentation have been described section 2.2.”

L143: what is meant by “membrane-bound peroxisomes.” All peroxisomes have a membrane, they are organelles…

L148: “K. marxianus is a GRAS” : delete ‘a’

L150: nonethanol: there is a hyphen. Use it wisely.

I do object to K. marxianus as a promising probiotic. This is far fetched and other yeasts work much better for this.

L165+: episomal plasmids are useless for metabolic engineering (at least in the final strain). They need to be maintained by selective pressure. This is costly and not efficient. Also the distribution of plasmids in a population is an issue.

Contrary to what the authors state: plasmids are not stable. There is a high loss-rate.

L172: “lower clonal variability”: how can that be? The copy numbers vary far more greatly compared to genomic integration.

L173+: “a centromeric plasmid was engineered to 173 regulate gene expression levels in Y. lipolytica”

A centromere may regulate copy number of a plasmid in a cell. Expression levels are regulated by promoters.

L175: there are potent expression systems…for this no research is required. TEF-promoters are everywhere.

L179+: this is simply wrong: these things have not “been widely used for the genetic engineering of nonconventional yeasts” 

L187: do the authors understand how the Cre-recombinase actually works? Inversion of the target sequence is not the goal…

L192: TALEN: please read up on this technology. Apparently, the authors do not know how this works.

L245: “excision of target loci” is not what Cas9 does. Cas9 generates a double strand break.

L247: “Considering the greatest aspect of the CRISPR–Cas9 system is its ability to edit the 247 genome accurately,” Really? The system simply generates a DSB. Accurate editing requires other tools.

L261: if you mention NHEJ-mutants you should also mention RAD51 overexpression as a way to enhance HR.

The section 4.3. reads better.

Author Response

Reviewer #2

Review on: Nonconventional yeasts engineered using the CRISPR–Cas system as emerging microbial cell factories

The review describes the utilization of CRISPR/Cas9 in non-conventional yeasts and catalogs the previously undertaken efforts.

As such this is a nice overview.

Response: Thanks a lot for the reviewer’s positive comment on our manuscript.

Comment 1           

It lacks a bit the description of various Cas9 enzymes and their PAM sequences as there is a larger diversity that could be used to also overcome off-target effects.

Response: Various Cas9 enzymes and their PAM sequences were described as a strategy to overcome off-target effects.

Comment 2

The authors conclude that there is an obstacle in NHEJ and CRISPR use. Maybe a look at some papers describing the overexpression of RAD51 to circumvent this issue would help in brightening up their outlook.

Response: Thanks a lot for the reviewer’s good point. We have thoroughly addressed this issue. Please, see the response to reviewer #1’s Comment 27.  

Comment 3

“Moreover, a superior defensive system against multiple stress factors can extend yeasts’ utility.” - There is no such thing as a “superior defense system”- rephrase and include specific examples e.g. high tolerance against salt, pH, temperature, osmotic stress etc.

Response: According to the reviewer, the sentence was rephrased.

Page 2, line 55-56: Moreover, a high tolerance against multiple stress factors such as heat, low pH, and salt can extend yeasts’ utility.

Comment 4

L57-59: I guess there are genome sequences for all these NCY. Otherwise CRISPR guide RNAs would be difficult to generate. Also these yeasts will all have safety level 1 status - even though they may not have been assigned GRAS. Actually, it is not challenging to develop suitable gene editing tools. What is challenging is to establish transformation protocols and selectable marker gnees to get going in the first place.

Response: Thanks a lot for the reviewer’s careful point. By this paragraph in Introduction, we intend to describe the current challenges of “general” NCYs (not limited to 5 candidate NCYs) associated with genetic engineering technologies “in comparison with” the conventional S. cerevisiae to emphasize the importance of NCYs using as platform. (Please, see expressions like “most”) The issues raised by the reviewer (i.e. genome sequence, safety level, challenges in gene editing tools) is reasonable; however, what we described on NCYs here is generally considered true. Nevertheless, we have revised manuscript for more clarity and to reflect the reviewer’s comment. We hope this is acceptable.

Page 2, line 58-63: Despite their beneficial traits, genetic information and manipulation tools for many NCYs are lacking compared with those for S. cerevisiae [11]. As most organisms have not been thoroughly analyzed for safety and lack genome sequencing data with the limited information on the exact gene loci and protein functions, it is challenging to develop and apply suitable gene editing tools, together with to establish transformation protocols and selectable marker genes [11].

Comment 5

L66-67: what is meant by traditional methods? TALEN? Zn-Finger? Or simple homologous recombination? Be specific.

Comment 6

L71: use Non-conventional yeast once and then abbreviate as NCY. Please change through-out the manuscript.

Response: According to the reviewer, “non-conventional yeast” was all abbreviated as NCY after its first appearance throughout the manuscript.

Comment 7

Fig 1: tolerant; modify ‘strong heat resistance’ - this may apply to archaea, certainly not to yeasts.

Response: ‘strong heat resistance” was changed to “heat tolerant” (See, revised Fig. 1)

Comment 8

2.1. rephrase ‘with a peroxisome system’ what is meant?

Comment 9

L113: it is saccharification

Comment 10

L125: non-pathogenic. And no: non-pathogenic does not equal GRAS. Be clear about the regulatory status.

Comment 11

L136+: S.cerevisiae may be incapable of xylose fermentation. Yet, there are 2nd generation biofuel yeasts that can do this job. Change and add appropriate references.

Response: Thanks a lot for the good point. Although there are other 2nd generation biofuel yeasts (either native or engineered) that can do xylose fermentation, this section intends to describe the industrial potential of “O. polymorpha” related to biofuel production in comparison with S. cerevisiae, the most widely used workhorse for bioethanol production. To address the issue by the reviewer and present our intention more specifically, the manuscript was revised with appropriate reference. 

Page 4, line 146-149: Because native S. cerevisiae, the most widely used workhorse for bioethanol production, is incapable of xylose fermentation and engineered S. cerevisiae to possess heterologous xy-lose metabolic pathway may suffer from high metabolic burden,

Comment 12

Delete: “The advantages of high-temperature fermentation have been described section 2.2.”

Response: Deleted.

Comment 13

L143: what is meant by “membrane-bound peroxisomes.” All peroxisomes have a membrane, they are organelles…

Page 4, line 154-155: methanol present within membrane-bound peroxisomes, which enables a compart-mentalized reaction.

Comment 14

L148: “K. marxianus is a GRAS” : delete ‘a’

Response: Deleted.

Comment 15

L150: nonethanol: there is a hyphen. Use it wisely.

Response: The hyphen was deleted.

Comment 16

I do object to K. marxianus as a promising probiotic. This is far fetched and other yeasts work much better for this.

Response: In agreement with the reviewer, the related expression has been toned down by deleting “promising”.

Comment 17

L165+: episomal plasmids are useless for metabolic engineering (at least in the final strain). They need to be maintained by selective pressure. This is costly and not efficient. Also the distribution of plasmids in a population is an issue.

Response: We agree with the reviewer’s opinion that episomal plasmids have several drawbacks as mentioned and genome integration is the preferred approach. However, there are also advantageous features over integrative or centromeric vectors, such as high copy number, high transformation efficiency, and overcoming genetic instability in multi-copy strains and non-specific genomic integration (frequently observed in genome integration), etc. For this reason, a variety of episomal plasmid expression system has been also developed for yeasts. In this context, we kindly ask the reviewer to keep the sentence as a way to demonstrate the current methodological status. Instead, the drawbacks of episomal plasmid-based expression were added in the revised manuscript.

Page 5, line 187-189: However, such an episomal plasmid system suffers from a high cost and inefficiency arising from segregational instability, which requires its maintenance under selective pressure for stable protein expression.

Comment 18

Contrary to what the authors state: plasmids are not stable. There is a high loss-rate.

Page 4, line 179-181: For S. cerevisiae, episomal vector systems with high copy numbers have been compartmentalized reaction through extensive optimization and are widely utilized relative to NCYs.

Comment 19

L172: “lower clonal variability”: how can that be? The copy numbers vary far more greatly compared to genomic integration.

Comment 20

L173+: “a centromeric plasmid was engineered to regulate gene expression levels in Y. lipolytica” A centromere may regulate copy number of a plasmid in a cell. Expression levels are regulated by promoters.

Comment 21

L175: there are potent expression systems…for this no research is required. TEF-promoters are everywhere.

Comment 22

L179+: this is simply wrong: these things have not “been widely used for the genetic engineering of nonconventional yeasts” 

Response: In accordance with the reviewer’s point, the sentence was toned down.

Page 5, line 198-199: Nuclease-based tools such as ZFNs and TALENs, can be used for the genetic engineering of NCYs [57,58]. However, these methods are outdated with the emergence of CRISPR–Cas9 system.

Comment 23

L187: do the authors understand how the Cre-recombinase actually works? Inversion of the target sequence is not the goal…

Response: The description of the Cre-loxP system has been improved based on the reviewer’s comment.

Page 5, line 200-204: The Cre–lox system, which deletes the sequences between direct repeats of two loxP sites by catalyzing their recombination, leaving a single loxP site behind in the genome [59], is also a useful technology for producing knockout mutants without leaving a marker; how-ever, some scars are left in the genome, or unwanted recombination may occur [10].

Comment 24

L192: TALEN: please read up on this technology. Apparently, the authors do not know how this works.

Response: Thanks a lot for the reviewer’s point. Following the reviewer 1’s request to simplify the explanation for ZFN and TALEN, the relevant text was removed.

Comment 25

L245: “excision of target loci” is not what Cas9 does. Cas9 generates a double strand break.

Comment 26

L247: “Considering the greatest aspect of the CRISPR–Cas9 system is its ability to edit the 247 genome accurately,” Really? The system simply generates a DSB. Accurate editing requires other tools.

Response: We agree with the reviewer’s point that strictly speaking, CRISPR–Cas9 system is not for genome editing but for generating DSB. Furthermore, following the reviewer 1’s comment on its redundancy, the sentence was removed in the revised manuscript. Please, see the response to Comment 25.

Comment 27

L261: if you mention NHEJ-mutants you should also mention RAD51 overexpression as a way to enhance HR.

Comment 28

The section 4.3. reads better.

Response: Thank you very much.

Round 2

Reviewer 1 Report

The authors have taken into account all relevant comments and improved the manuscript sufficiently.

Reviewer 2 Report

revision done ok